# Contributions of Net Charge on the PlyC Endolysin CHAP Domain

**DOI:** 10.3390/antibiotics8020070

**Published:** 2019-05-28

**Authors:** Xiaoran Shang, Daniel C. Nelson

**Affiliations:** 1Institute for Bioscience and Biotechnology Research, Rockville, MD 20850, USA; sxr520@umd.edu; 2Department of Veterinary Medicine, University of Maryland, College Park, MD 20742, USA

**Keywords:** endolysin, PlyC CHAP, protein net charge, CBD-independent, FoldX

## Abstract

Bacteriophage endolysins, enzymes that degrade the bacterial peptidoglycan (PG), have gained an increasing interest as alternative antimicrobial agents, due to their ability to kill antibiotic resistant pathogens efficiently when applied externally as purified proteins. Typical endolysins derived from bacteriophage that infect Gram-positive hosts consist of an N-terminal enzymatically-active domain (EAD) that cleaves covalent bonds in the PG, and a C-terminal cell-binding domain (CBD) that recognizes specific ligands on the surface of the PG. Although CBDs are usually essential for the EADs to access the PG substrate, some EADs possess activity in the absence of CBDs, and a few even display better activity profiles or an extended host spectrum than the full-length endolysin. A current hypothesis suggests a net positive charge on the EAD enables it to reach the negatively charged bacterial surface via ionic interactions in the absence of a CBD. Here, we used the PlyC CHAP domain as a model EAD to further test the hypothesis. We mutated negatively charged surface amino acids of the CHAP domain that are not involved in structured regions to neutral or positively charged amino acids in order to increase the net charge from -3 to a range from +1 to +7. The seven mutant candidates were successfully expressed and purified as soluble proteins. Contrary to the current hypothesis, none of the mutants were more active than wild-type CHAP. Analysis of electrostatic surface potential implies that the surface charge distribution may affect the activity of a positively charged EAD. Thus, we suggest that while charge should continue to be considered for future engineering efforts, it should not be the sole focus of such engineering efforts.

## 1. Introduction

Bacteriophage (phage) endolysins are peptidoglycan (PG) hydrolases produced by phage at the end of a lytic cycle [1]. In the presence of holins, pore-forming proteins, endolysins can pass the cytoplasmic membrane to degrade the PG layer of the cell wall, resulting in the lysis of the bacteria and release of new progeny virions [2]. These enzymes are also capable of destroying the Gram-positive bacterial PG from outside the cell as recombinant proteins [3]. Due to the physical barrier of the outer membrane, exogenously added endolysins usually cannot access the PG of Gram-negative bacteria. However, engineered endolysins with cationic or membrane-disrupting peptides have been reported to successfully kill Gram-negative bacteria “from without” [4]. Consequently, endolysins are novel antimicrobial agents and can be used to treat both Gram-positive and Gram-negative antibiotic-resistant bacterial infections because their mode of action is not inhibited by traditional resistance mechanisms [3]. 

Endolysins derived from phage that infect Gram-positive hosts have very similar modular structures with one or more N-terminal enzymatically-active domains (EADs) and a C-terminal cell wall binding domain (CBD) [5]. The EADs that cleave covalent bonds in the PG are conserved into five mechanistic classes: muramidases, glucosaminidases, N-acetylmuramyl-L-alanine amidases, endopeptidase, and lytic transglycosylases. In contrast, the CBDs possess no enzymatic activity but rather function to bind to specific ligands on the cell wall, which are usually secondary wall carbohydrates or teichoic acid moieties. Thus, the endolysin host range is often dictated by the specificity of the CBD, which is either broad-spectrum, targeting molecules harbored by a bacterial genus or multiple genera, or narrow-spectrum, targeting molecules shared by a single species or serovar [6,7]. The CBDs have been shown to be essential for function of EADs in a number of modular endolysins, including PlyGRCS [8], PlySs2 [9], PlyB [10], Cpl-1 [11], and PlyB30 [12]. 

Whereas many EADs require the presence of the CBD for binding and subsequent activity, some EADs can bind the bacterial surface independently of the CBD, and a few even have increased enzymatic activity compared to the full-length endolysin. One example is the staphylococcal phage endolysin, LysK. The LysK EAD, a cysteine-histidine amidohydrolase/endopeptidase (CHAP), alone displays higher lytic activity against staphylococci than the full-length LysK [13]. Similarly, when the Group B streptococcal phage endolysin, PlyGBS, was truncated to remove the CBD from the EAD, a ~20 fold increase in specific activity was noted compared to PlyGBS [14]. Moreover, without the constraining binding properties of the CBD, some EADs from modular endolysins show an extended host range compared to their parental full-length endolysins. Examples include the EAD of the *Bacillus anthracis* phage endolysin, PlyL [15] and the EAD of the *Clostridium difficile* phage endolysin, CD27L [16].

The reason(s) as to why some EADs can target and lyse the PG in the absence of a CBD, whereas the presence of a CBD is an absolute requirement for activity in other EADs, is unknown. However, a thought-provoking study by Low et al. [15] suggested that a net positive charge of an EAD enables it to function independently of its CBD, presumably through ionic interactions with the bacterial surface, which characteristically has a net negative charge due to surface carbohydrates. This conceptual understanding was then applied by the authors to endolysin bioengineering studies. For example, the EAD of a *Bacillus subtilis* prophage endolysin, XlyA, had a net charge (Z) of -3 at neutral pH and displayed no lytic activity against *B. subtilis* cells in the absence of its CBD. Site-directed mutagenesis of five non-cationic residues to lysine (K) produced a shift in net charge from Z = −3 to Z = +3, and the mutated XlyA EAD alone was able to lyse *B. subtilis* cells at a rate nearly identical to that of full-length XlyA. In a separate study, the addition of a positively-charged peptide enhanced the lytic activity of the λSa2lys endolysin [17], suggesting the positive charges may increase the avidity of the enzyme for the bacterial surface. 

In the present work, we sought to validate Low’s hypothesis. The model EAD for this study is the CHAP domain from the PlyC endolysin [18]. This EAD possesses potent catalytic activity and is amenable to engineering, as it has been subjected to mutational analysis to improve thermostability [19] and has been used as the EAD in chimeragenesis projects incorporating different CBDs (i.e., ClyR [20] and ClyJ [21]). The homolog of the PlyC CHAP domain via a structural DALI search is the LysK CHAP domain, which is known to harbor improved activity compared to full-length LysK. However, the PlyC CHAP catalytic domain, in contrast, loses most (~99%) lytic activity in the absence of PlyCB (i.e., the CBD of PlyC) [18]. The net charge of the LysK CHAP is Z = +1, whereas the net charge of the PlyC CHAP is Z = −3. Therefore, the PlyC CHAP is a good candidate to test Low’s hypothesis, proposing that a conversion of net charge on an EAD will enable it to display lytic activity in the absence of a CBD.

## 2. Results

### 2.1. Library of PlyC CHAP Mutants

The PlyC CHAP domain (i.e., the C-terminal EAD of PlyCA comprising amino acid 309-465) was isolated from the PlyC holoenzyme crystal structure and edited in PyMOL (atomic coordinates were only available for amino acid 310-464). Five surface and unstructured residues, Asp-311, Asp-355, Asp-363, Asp-429, and Asp-450, were selected for mutagenesis (Figure 1). Through different combinations of point mutations incorporating either a neutral charge (i.e., alanine) or a positive charge (i.e., lysine) in place of each aspartic acid residue, a library of 192 mutant candidates harboring a net charge between +1 and +7 was computationally generated.

### 2.2. Prediction of the Properly Folded PlyC CHAP Mutants via ΔΔG_FoldX_

FoldX is a computational biology tool developed for rapid evaluation of the effect of mutations on stability, folding, and protein dynamics [22]. FoldX was used to narrow down the 192 mutants via a change in free energy of the mutant relative to the wild-type (WT) protein (ΔΔG_FoldX_ = ΔG_mut_ – ΔG_WT_). A negative ΔΔG_FoldX_ (ΔΔG_FoldX_ < 0) suggests that the mutation is more stable than the WT protein and should fold properly. However, 79% of the mutants were predicted to have a positive ΔΔG_FoldX_, meaning these mutations had destabilizing effects (Figure 2). At each charge category (Z = +1 to Z = +7), only the mutants possessing the largest predicted negative ΔΔG_FoldX_ were chosen to be made (Table 1). Notably, the selected +6 charged and +7 charged CHAP mutants contained either neutral or positive ΔΔG_Foldx_, probably due to the high number of required mutations (Table 1).

### 2.3. Protein Solubility and Purity

All chosen mutants were expressed and purified by nickel affinity chromatography. The 6x His-tag at the N-terminus of each protein, which might affect the net surface charge in solution, was cleaved at an engineered thrombin cleavage site before further purification. The SDS-PAGE gel after His-tag removal suggested that PlyC CHAP mutants were pure and as soluble as WT CHAP (Figure 3). PlyC CHAP+7 displayed as a double band in the SDS-PAGE may indicate degradation of the protein since the positive ΔΔG_FoldX_ (Table 1) suggests this mutant is slightly unstable. 

### 2.4. In Vitro PlyC CHAP Mutants’ Activity

PlyC is one of the most potent endolysins studied to date [23] and the PlyC CHAP domain is known to require its CBD for full activity. However, despite the Z = −1 charge, the PlyC CHAP domain does retain a very small (<1% of PlyC), but measurable and reproducible lytic activity against sensitive streptococcal species [18]. A turbidity reduction assay was used to benchmark the lytic activity of PlyC CHAP mutants to WT PlyC CHAP. However, none of the CHAP mutants displayed increased lytic activity compared with WT using *Streptococcus pyogenes* D471 as host over a broad concentration range (Figure 4). Nonetheless, the data revealed several interesting aspects. First, the CHAP mutants with net +1 and +2 surface charges (i.e., CHAP+1 and CHAP+2) showed the same lytic activity as WT CHAP, which suggests that the positive charge alone does not affect lytic activity. Second, in the low concentration ranges (< 16 μg/mL), the CHAP mutants with +3 to +7 surface charges (CHAP+3 to CHAP+7) were virtually devoid of lytic activity, but as the concentration increased, they had similar lytic activity to WT CHAP as well as CHAP+1 and CHAP+2. The activity noted for CHAP WT compared to the full PlyC holoenzyme is consistent with previous data [18], and presumably represents activity resulting from random collisions of CHAP with the cell wall in the absence of the PlyC CBD. 

### 2.5. Analysis of PlyC CHAP Electrostatic Surface Potential

The surface charge distributions were then examined through CCP4MG software [24]. The active-site residues (C333 and H420) of the PlyC CHAP are located in a neutral groove, which remains unchanged in CHAP mutants (Figure 5A, neutral groove on far right column). Although an overall increased surface charge indicates an increased positive electrostatic potential in the CHAP mutants, the regions accumulating the positive surface potential are evenly distributed over the entire CHAP surface (Figure 5A). Low et al. [15] did not imply any relationship between the relative position of the active-site and the distribution of positive charge in their work. However, when we examined the surface potential of their enzyme, XlyA, and its mutant, XlyA+5K, we did notice an accumulation of positive surface potential near the negative charged active-site (Figure 5B, negative groove on far right column). Thus, a simple conversion of the surface charge on an EAD may not be adequate by itself to create CBD-independent lytic activity.

## 3. Discussion

Bacteriophage have optimized endolysins for lytic activity at high concentrations from within the bacterial intracellular environment through coevolution with bacterial hosts to ensure phage release and survival. In contrast, these enzymes have not evolved to be used exogenously to produce lysis “from without” and, therefore, may not be fully optimized for this application. As such, there exists an engineering potential for these enzymes to be modified to increase their activity, alter host range, or overcome complex extracellular environments [25]. As a growing amount of research focuses on the modular design and crystal structures of endolysins, structure-based rational engineering approaches, such as chimeragenesis and structure-guided mutagenesis, can be used to produce engineered endolysins with desirable antimicrobial properties [26].

Chimeragenesis is a method used to exchange functional modules (i.e., EADs and CBDs) for better activities and/or an altered or expanded host range. This engineering approach has been exploited by nature itself through horizontal gene transfer, such as the pneumococcal endolysin Pal whose EAD is most similar to the endolysin of *Lactococcus lactis* phage BK5-T, whereas its CBD is homologous to other pneumococcal endolysins [27]. A well-studied chimeric endolysin that extends the host range of the parental enzymes is ClyR [20,28]. This enzyme is a fusion of a PlyC CHAP EAD and the PlySs2 CBD. While each parental enzyme has a narrow host range to select streptococcal species, the ClyR chimera possesses a broad activity against almost all streptococcal species as well as staphylococcal and enterococcal species. Other engineered chimeras, including ClyS [29], Cpl-711 [30], Csl2 [31], and PL3 [32] result in endolysins that displayed enhanced antimicrobial activity over either parental enzyme. Some chimeric endolysin, such as λSA2-E-Lyso-SH3b and λSA2-E-LysK-SH3b include multiple EADs from different endolysins and result in both extended host range and increased antimicrobial activity [33,34].

Structure-guided mutagenesis is expected to expand bioengineering efforts toward modulating endolysin properties as more high resolution X-ray crystallography and NMR structures are solved. The atomic coordinates can be used for computational modeling to predict mutations that can affect enzymatic activity, substrate binding affinity, and interactions with the solvent interface or intra- or inter-domain interactions that may affect stability of the enzyme. Notably, several algorithms, such as Rosetta [35] and FoldX [22,36], can be used to rapidly assess point mutations *in silico* and calculate the effect these mutations have on protein stability through the approximation of Gibbs free energy (Δ*G*). The top resulting mutants can then be made and characterized experimentally. Such an approach was recently done to introduce a thermostabilizing mutation, T406R, to the EAD of PlyC. This point mutant introduced a critical hydrogen bond between two domains and resulted in a 16 fold increase of half-life at 45 °C [19]. 

Low et al. [15] used similar site-directed mutagenesis techniques to introduce five lysine residues to non-structured, non-catalytic regions of the XlyA endolysin EAD, effectively changing the charge from Z = -3 to Z = +3. Whereas the XlyA EAD had no lytic activity against *B. subtilis* in the absence of its CBD, the mutant EAD, XlyA+5K, possessed WT XlyA activity independent of the CBD. The authors postulated that the net positive charge allowed for interaction with the “continuum of negative charge” on the secondary cell wall polymers on the bacterial surface, thus obviating the need for a CBD. To contrast this gain-of-function experiment, the authors also tested a loss-of-function engineering approach using the PlyBa04 EAD, which carries a net charge of Z = +1. Unlike the native XlyA EAD, the PlyBa04 EAD is able to lyse its target, *B. cereus*, as efficiently as the full-length endolysin. However, after site-directed mutagenesis to change the net charge from Z = +1 to Z = −3, the resulting PlyBa04 EAD mutant showed negligible lytic activity towards *B. cereus*. The authors concluded that engineering a reversal of net charge on an EAD could be used to either create or eliminate CBD dependence, which could be used in future bioengineering studies to fine-tune endolysin activity. Two years later, a separate study by Diez-Martinez et al. [37] with the Cpl-7 pneumococcal endolysin supported Low’s charge hypothesis. These authors introduced charge substitutions to 15 amino acids of the Cpl-7 CBD resulting in a change from Z = −14.93 to Z = +3.0 at neutral pH. The mutant, named Cpl-7S, displayed significantly increased lytic activity against several streptococcal strains compared to WT Cpl-7. 

The ultimate goal of our study was to create an engineered endolysin that is simple (i.e., one catalytic domain) and works on a very broad host range (i.e., does not require a CBD, meaning the EAD alone defines host range). Toward this end, we sought to engineer the PlyC CHAP domain to be such an enzyme using engineering principles guided by the findings of Low et al. [15]. Toward this end, we successfully made a range of positively charged CHAP mutants. The crystal structure of PlyC provided a model for selecting the potential point mutations. The computational tool, FoldX, helped narrow down the candidates from a total of 192 to 40. The ones that were most stable among the 40 candidates were used for cloning, protein expression, and purification. The achievement of this experimental design suggests that computational tools, like FoldX, can be used in the upstream evaluation providing a rationale for the selected mutations. 

Our results indicated that none of the positively charged CHAP mutants displayed higher lytic activity than WT CHAP. Thus, at least for the PlyC CHAP, the hypothesis developed by Low et al. [15] is not supported. Nonetheless, while our approach was similar to Low et al. [15] in that mutations were rationally selected based on being solvent/surface accessible, did not participate in hydrogen bonding or salt bridges, and were not part of ordered structures, our mutations were globally distributed over the surface of the PlyC CHAP whereas Low’s mutations were more tightly grouped around the XlyA active-site. Additionally, we note that the PlyC CHAP active-site is located along a neutrally charged groove and the XlyA active-site is within an anionic pocket. It is presently not known if these differences account for the opposing results of our work with those of Low et al. [15]. We suggest that while the charge hypothesis should continue to be acknowledged for future engineering efforts, it should not be the sole focus and other characteristics (i.e., surface charge distribution, nature of active-site electrostatics, etc.) should also be taken into consideration.

## 4. Materials and Methods 

### 4.1. Bacterial Strains and Culture Conditions

*Streptococcus pyogenes* D471 was cultured from a −80 °C frozen stock and grown in Todd Hewitt broth supplemented with 1% yeast extract (THY) without shaking at 37 °C. *E. coli* strains DH5α and BL21 (DE3) were grown in Luria-Bertani (LB) broth. When needed, kanamycin (50 μg/mL) was added to the media. All bacterial cultures were grown at 37 °C in a shaking incubator unless otherwise stated. 

### 4.2. In Silico Modeling of PlyC CHAP Mutants

The crystal structure coordinates were obtained from the Protein Data Bank for PlyC (4F88), XlyA (3RDR), and XlyA+5K (3HMB). The strategy used to change the net charge (Z) of the CHAP domain was to substitute negatively charged amino acids, aspartic acid (D) and glutamic acid (E), that were surface exposed and not involved in structured regions (i.e., α-helix or β-sheet) to neutral (alanine (A)) or positively charged (lysine (K)) amino acids to increase the Z score from −3 to +1 through +7 at pH 7.4. The net charges were calculated from the online Protein Calculator Version 3.4 [38]. PyMOL (The PyMOL Molecular Graphics System, Version 1.7.4 Schrödinger, LLC) was used to identify surface amino acids and a library of 192 CHAP mutant candidates was established following the outlined strategy. Before further validation, the CHAP mutant candidates had their side-chain orientation optimized using the FoldX 3.0 Repair PBD command [36]. The resulting coordinates were then processed by FoldX 3.0 for calculating the free energy change of the mutants (ΔΔG_FoldX_). The desirable mutants possessed ΔΔG_FoldX_ < 0 kcal/mol (ΔΔG_FoldX_ = ΔG_mut_ – ΔG_WT_) and the mutants with the largest negative ΔΔG_FoldX_ were then picked for experimental study. The electrostatic surface potential was imaged using CCP4MG [24].

### 4.3. Cloning and Site-Directed Mutagenesis

The primers used in this study are listed in Table 2. The WT gene of PlyCA CHAP domain was amplified from pBAD24::*plyC* [39] and cloned via NdeI and BamHI sites into pET28a, as the template for the mutagenesis. The Change-IT^TM^ Multiple Mutation Site-Directed Mutagenesis Kit from Affymetrix was used to generate all mutants. Each mutation was designed to be in the middle of a 30 nucleotide phosphorylated forward primer and the mutagenesis followed instructions provided by the manufacturer of the kit. The resulting mutants were confirmed by sequencing (Macrogen, Rockville, MD, USA) before being transformed into *E. coli* BL21 (DE3) for protein expression.

### 4.4. Protein Expression and Purification

The overnight cultures of *E. coli* BL21 (DE3) harboring the WT PlyCA CHAP domain or mutants were sub-cultured 1:100 into 1.5 L LB supplemented with kanamycin in a 4 L baffled Erlenmeyer flask at 37 °C in a shaking incubator. The culture was induced at mid-log phase (about 4 h) with 1 mM isopropyl β-D-1-thiogalactopyranoside (IPTG) and incubated at 18 °C overnight. The next morning, *E. coli* BL21 (DE3) bacterial cells were harvested at 5,000 rpm, resuspended in PBS, pH 7.4, sonicated, and clarified via centrifugation at 12,000 rpm at 4 °C. The soluble portion of the cell lysate was applied to a Ni-NTA resin column (Thermo Fisher Scientific, Waltham, MA, USA). The 6X His-tagged protein was washed and eluted using a gradient of imidazole from 20 mM to 500 mM in PBS buffer, pH 7.4. The protein purity was assessed on a 7.5% SDS-PAGE gel before dialysis to remove the imidazole. The 6x His-tag was removed using the Thrombin Cleavage Capture Kit (EMD Millipore, Burlington, MA, USA) according to the protocol provided by the manufacturer.

### 4.5. In Vitro PlyC CHAP Activity

The activities of the PlyC CHAP domain and its mutants were evaluated via a spectrophotometric-based turbidity reduction assay as described previously [6]. An overnight culture of *S. pyogenes* D471 was harvested at 4500 rpm for 10 min, washed twice and resuspended in PBS, pH 7.4 buffer to reach an OD_600_ = 2.0. In a flat-bottom 96-well plate (Thermo Fisher Scientific), bacterial cells were mixed 1:1 with equimolar amounts of the PlyC CHAP domain or its mutants and the OD_600_ was monitored every 15 sec for 1 hour at 37 °C using a SpectraMax 190 spectrophotometer (Molecular Devices, San Jose, CA, USA). The OD_600_ decrease was calculated after 1 hour treatment by calculating the difference in OD_600_ readings between the PBS treated bacteria and the enzyme treated bacteria. Each assay was conducted in triplicate.

## 5. Conclusions

Bacteriophage-encoded endolysins offer an emerging alternative to conventional antibiotics. These enzymes function to degrade the bacterial peptidoglycan, producing fatal osmotic lysis in susceptible Gram-positive bacteria. A cell wall binding domain (CBD) is often, but not always, required for activity. In some endolysins, the catalytic domain alone is sufficient for lysis. In these examples, it has been noted that the catalytic domains have a net positive charge, suggesting they can interact with the negatively charged bacterial surface in the absence of a CBD. Protein engineering studies support this hypothesis by showing that a change in net charge of a catalytic domain, from negative to positive, correlated with CBD independence. However, similar experiments on the PlyC endolysin catalytic domain reported here do not support the hypothesis. While not discounting the charge hypothesis, other factors, such as distribution of surface charges, may also affect the CBD dependence of an endolysin and should be considered for future protein engineering efforts. 

## Figures and Tables

**Figure 1 antibiotics-08-00070-f001:**
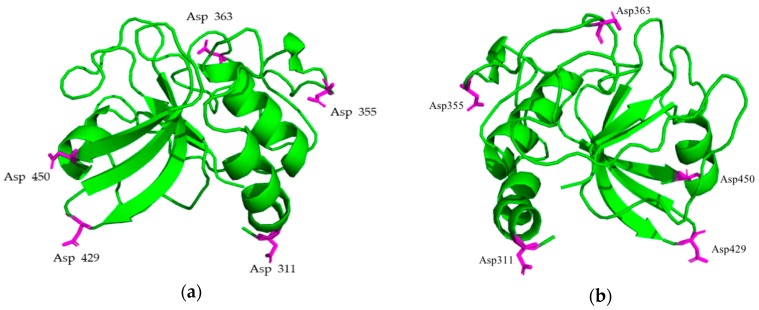
Mutation sites of PlyC CHAP. (**a**) 3.3 Å resolution of the PlyC CHAP crystal structure. The magenta-colored amino acids represent the potential mutation sites. (**b**) 180° horizontal rotation of (a). The mutation sites are solvent exposed, not structured in α-helix or β-sheets, and form no interactions with other residues.

**Figure 2 antibiotics-08-00070-f002:**
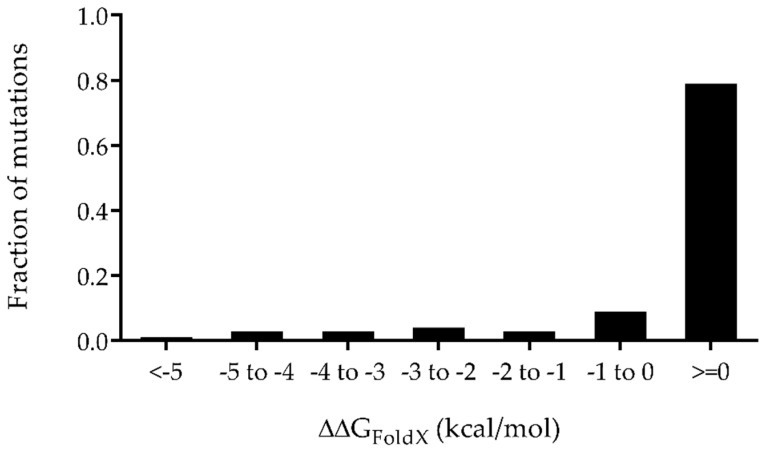
Distribution of the predicted change in folding free energy (∆∆G_foldX_) for all 192 possible CHAP mutants calculated with FoldX3.0. Mutations with ∆∆G_foldX_ < 0 are expected to retain the same folding characteristics as WT CHAP.

**Figure 3 antibiotics-08-00070-f003:**
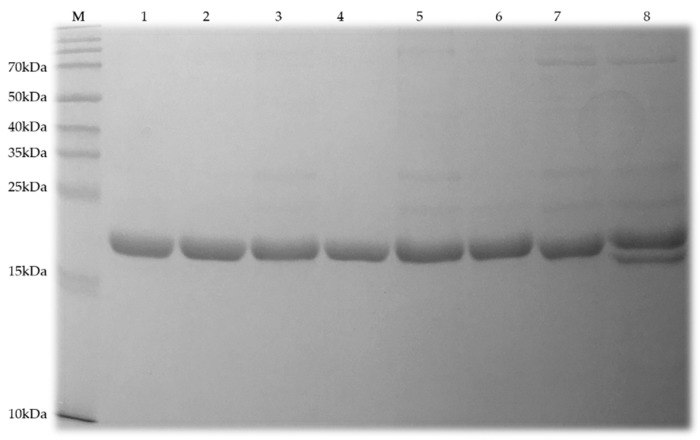
SDS-PAGE analysis of WT PlyC CHAP and its mutants. The solubility and purity of each enzyme after the His-tag cleavage were assessed via a 7.5% SDS-PAGE gel. The lanes correlate to: (M) BioRad protein markers; (**1**) PlyC CHAP WT; (**2**) PlyC CHAP +1; (**3**) PlyC CHAP +2; (**4**) PlyC CHAP +3; (**5**) PlyC CHAP +4; (**6**) PlyC CHAP +5; (**7**) PlyC CHAP +6, and; (**8**) PlyC CHAP +7.

**Figure 4 antibiotics-08-00070-f004:**
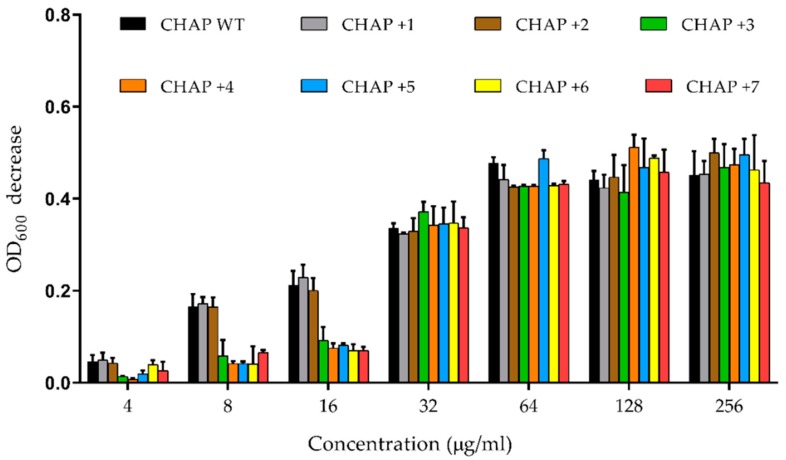
In vitro lytic activity against *S. pyogenes* D471. The different concentrations of enzymes were added to the overnight bacterial cultures. The OD_600_ was recorded every 15 seconds for 1 hour. The OD_600_ decrease, which is the net change in OD_600_ between the PBS treated control and enzyme treatment for 1 hour, was represented as the enzyme activity. The experiment was conducted in triplicate, and the error bars represent the standard deviation.

**Figure 5 antibiotics-08-00070-f005:**
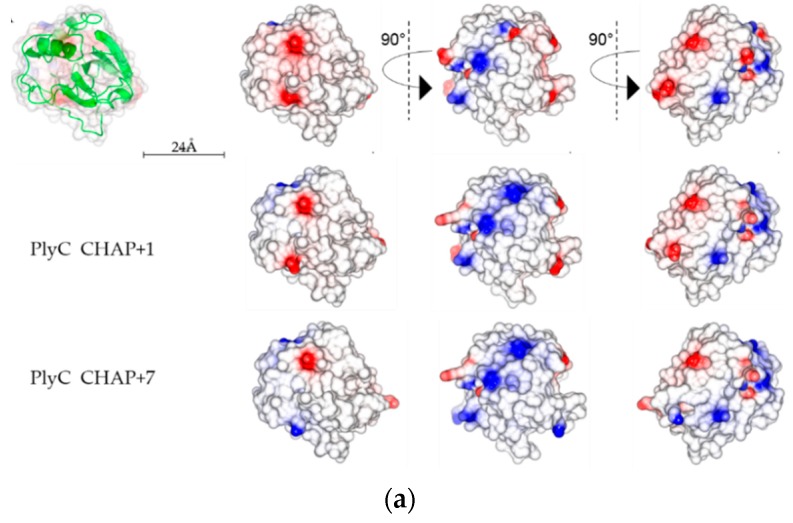
CCP4MG generated electrostatic surface potential maps of PlyC CHAP, XlyA, and their mutants. Surfaces are color-coded according to electrostatic potential (calculated by the Poisson-Boltzmann solver within CCP4MG). The color of the surface represents the electrostatic potential on the protein surface, going from blue (potential of +10kT/e) to red (potential of -10kT/e). (**a**) Electrostatic surface potential of PlyC CHAP WT, CHAP +1, and CHAP +7 in different orientations. The active-site of PlyC CHAP is in a neutral groove observed in the far right column. (**b**) Electrostatic surface potential of XlyA and XlyA+5K in different orientations. The active-site of XlyA is in a negative groove observed in the far right column.

**Table 1 antibiotics-08-00070-t001:** PlyCA CHAP Selected Mutations.

Mutant Name	pI (Isoelectric Point)	Net Charge (Z) at pH 7.4	Point Mutations	ΔΔG_FoldX_ (kcal/mol) = ΔG_Mut_-ΔG_WT_
CHAP WT	6.11	−3	N/A	0
CHAP +1	7.89	+1	D311K:D355K	−5.32
CHAP +2	8.29	+2	D311K:D355K:D429A	−4.61
CHAP +3	8.59	+3	D311K:D355K:D363K	−4.32
CHAP +4	8.88	+4	D311K:D355K:D363K:D429A	−4.13
CHAP +5	9.11	+5	D311K:D355K:D363K:D429K	−2.49
CHAP +6	9.30	+6	D311K:D355K:D363K:D429A:D450K	0.01
CHAP +7	9.43	+7	D311K:D355K:D363K:D429K:D450K	1.11

**Table 2 antibiotics-08-00070-t002:** Primer information.

Plasmid	Template	Primer	Sequence
CHAP D311K	pET28a::*chap*	XS3	5′-ATGGGGTCTAAAAGAGTTGCAGCAAAC-3′
CHAP D355K	pET28a::*chap*	XS4	5′-TCATACTCAACAGGTAAACCAATGCTACCGTTA-3′
CHAP D363K	pET28a::*chap*	XS5	5′-CTACCGTTAATTGGTAAAGGTATGAACGCTCAT-3′
CHAP D429K	pET28a::*chap*	XS6	5′-ATTGAAAGCTGGTCAAAAACTACCGTTACAGTC-3′
CHAP D429A	pET28a::*chap*	XS8	5′-ATTGAAAGCTGGTCAGCGACTACCGTTACAGTC-3′
CHAP D450K	pET28a::*chap*	XS7	5′-ATACGCAGCACCTATAAACTTAACACATTCCTA-3′

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
