# Peer review of "Contributions of Net Charge on the PlyC Endolysin CHAP Domain"

_antibiotics, 2019, doi:10.3390/antibiotics8020070_

Round 1

Reviewer 1 Report

In the manuscript entitled “Contribution of net charge on the PlyC endolysin CHAP domain” Shang and colleagues report on the impact of net positive charge for the lytic activity of endolysin catalytic domains. It was previously reported that net charge modification enables it to work independently of its binding domain, which opens interesting possibilities for endolysin bioengineering. Indeed, catalytic domains were sometimes observed not to have lytic activity when expressed without their binding domain. This situation can be possibly improved by net charge modification.

A general comment is that the proposed manuscript is really well written and set up in a logical manner. In addition, experiments are well executed and the literature in the field is rigorously documented. However, a major comment would be on the reason why the authors decided to modify the net charges of the PlyC CHAP catalytic domain. Indeed, since it is already active on its own, what was the expectations of the authors by modifying its net charges?

Major comments:

Authors have to be more precise regarding the benefits of modifying the net charges of the CHAP domain.

Authors have to add the activity of the native PlyC lysin in figure 4.

In addition, authors make the statement at line 23-24 that none of the mutants were as active as the WT, which seems not to be true regarding figure 4, where CHAP +1 and +2 are as active as the WT. All mutants also have the same activity at higher concentration.

Minor comments:  

- Authors have to make a comment on the double band present in Figure 3, line 8. Is it the same protein?

- Figure 4. What is the OD decrease? Is it after 1h and compared to a control without lysin?

Author Response

Response to Reviewer 1 Comments

Reviewer 1

Comments and Suggestions for Authors

In the manuscript entitled “Contribution of net charge on the PlyC endolysin CHAP domain” Shang and colleagues report on the impact of net positive charge for the lytic activity of endolysin catalytic domains. It was previously reported that net charge modification enables it to work independently of its binding domain, which opens interesting possibilities for endolysin bioengineering. Indeed, catalytic domains were sometimes observed not to have lytic activity when expressed without their binding domain. This situation can be possibly improved by net charge modification.

A general comment is that the proposed manuscript is really well written and set up in a logical manner. In addition, experiments are well executed and the literature in the field is rigorously documented. However, a major comment would be on the reason why the authors decided to modify the net charges of the PlyC CHAP catalytic domain. Indeed, since it is already active on its own, what was the expectations of the authors by modifying its net charges?

Major comments:

Authors have to be more precise regarding the benefits of modifying the net charges of the CHAP domain.

The PlyC CHAP domain, which has a net negative charge, has very low activity (~1% WT) in the absence of its cell wall binding domain (CBD). In contrast, the LysK CHAP domain, which has a net positive charge, has high activity (~200-500% WT) in the absence of its CBD. Therefore, we reasoned that changing the net charge on the PlyC CHAP domain from negative to positive might impart a noticeable increase in activity in the absence of its CBD. We have added additional text to lines 83-84 of the introduction to further clarify this point.

Authors have to add the activity of the native PlyC lysin in figure 4.

We appreciate the comments. PlyC is one of the most active endolysins described given its synergy between two catalytic domains. While the CHAP domain has moderate activity by itself compared to other enzymes, it has less than 1% of the activity of native PlyC (refer to McGowan et al.). By addition of the WT PlyC activity to Figure 4, the scale will dramatically change and it will be difficult to visualize the differences between the CHAP domain and its mutants. Thus, we decided not to add the activity of PlyC to this figure, but instead we state its activity in lines 131-133 for reference.

Also, authors make the statement at line 23-24 that none of the mutants were as active as the WT, which seems not to be true regarding figure 4, where CHAP +1 and +2 are as active as the WT. All mutants also have the same activity at higher concentration.

We agree with the reviewer. We have restated lines 23-24 as “none of the mutants were more active than wild type CHAP”.

 Minor comments:  

- Authors have to make a comment on the double band present in Figure 3, line 8. Is it the same protein?

The upper band is in lane 8 is the size of our target protein and we believe the lower band is the breakdown product of it. The lower band co-eluted off the nickel column, so it retains the N-terminal 6X-His tag. Notably, the CHAP+7 mutant is the only mutant that possesses a positive ΔΔGFoldX, suggesting the protein is slightly unstable and therefore more amenable to degradation, which we believe is observed in lane 8. We have now added information to lines 123-124 to describe the double band in the gel.

- Figure 4. What is the OD decrease? Is it after 1h and compared to a control without lysin?

The OD600 decrease was calculated after 1-hour treatment. It was the final OD600 reading of the enzyme treatment subtracted from the PBS treated control. We have now added additional text to lines 149-150 and lines 300-301 to provide additional clarity.

Reviewer 2 Report

In the manuscript “Contributions of Net Charge on the PlyC Endolysin CHAP Domain” of Xiaoran Shang and Daniel C. Nelson, the authors reported on the use of plyC CHAP domain as a model A system to investigate if a net positive charge on the EAD enables it to reach the negatively charged bacterial surface via ionic interactions in the absence of a CBD. Therefore the authors changed the amino acid composition to increase the net charge. The generated mutants could be successfully expressed and purified. However, none of the mutants showed a comparable activity like the wild type protein.

This reviewer feels that this manuscript can be published without further revisions. All necessary data were provided. The manuscript is well written and comprises all experimental details.

There are no further comments of this reviewer. It is a really nice paper!

Author Response

 Reviewer 2

Comments and Suggestions for Authors

In the manuscript “Contributions of Net Charge on the PlyC Endolysin CHAP Domain” of Xiaoran Shang and Daniel C. Nelson, the authors reported on the use of plyC CHAP domain as a model A system to investigate if a net positive charge on the EAD enables it to reach the negatively charged bacterial surface via ionic interactions in the absence of a CBD. Therefore the authors changed the amino acid composition to increase the net charge. The generated mutants could be successfully expressed and purified. However, none of the mutants showed a comparable activity like the wild type protein.

This reviewer feels that this manuscript can be published without further revisions. All necessary data were provided. The manuscript is well written and comprises all experimental details.

There are no further comments of this reviewer. It is a really nice paper!

We thank the reviewer for the comments.

Round 2

Reviewer 1 Report

Authors significantly improved the overall quality of the manuscript and could address most of the critics.